# Effect of Mg–Ti Treatment on Nucleation Mechanism of TiN Inclusions and Ferrite

**Tianpeng Qu, Caiwei Zhang, Deyong Wang \*, Jie Zhan, Jun Tian and Dong Hou**

Shagang School of Iron and Steel, Soochow University, Suzhou 215021, Jiangsu, China; qutianpeng@suda.edu.cn (T.Q.); qibao0902@163.com (C.Z.); 13677983406@163.com (J.Z.); jtian@suda.edu.cn (J.T.); houdong0702@suda.edu.cn (D.H.)

\* Correspondence: dywang@suda.edu.cn; Tel.: +86-0512-6716-5621

**Abstract:** Large sizes of columnar crystals and TiN particles have a great influence on the surface quality of ferritic stainless steel. In the present paper, this study proposed to obtain fine-grained equiaxed structures by Mg–Ti treatment. Through the experiment happened in resistance furnace with argon protection, the refining effect of Mg–Ti addition on the microstructure and TiN particles were investigated, and the refinement mechanism was discussed from interface coherence theory. It was found that due to adding Mg and Ti into molten ferritic stainless steel, the equiaxed crystal ratio increased from 37% to 50%, and the size of TiN particles reduced at the same time. The lattice matching characteristics of $MgAl_2O_4$/TiN and TiN/$\delta$-Fe were investigated by FIB-HRTEM. According to Bramfitt's equation, the lattice misfit for (400)$MgAl_2O_4$||(200)TiN and (200)TiN||(110)$\delta$-Fe was 5.02% and 4.41%, respectively, which were all belong to the effective nucleation range. It could be considered that MgO and $MgAl_2O_4$ formed in the molten steel promoted TiN nucleation easier to precipitate out with large quantities in the liquid phase. The TiN particles with more uniform distribution significantly enhanced the heterogenous nucleation of ferritic phase during initial solidification process base on the good lattice fitting condition. Finally the equiaxed crystal ratio of $\delta$-Fe phase increased dramatically.

**Keywords:** Mg–Ti treatment; ferritic stainless steel; heterogeneous nucleation; lattice misfit

## 1. Introduction

Ferritic stainless steel, with chromium content from 15% to 30% (weight), has been widely used for its outstanding characteristics, such as lower thermal expansion coefficient, better resistance to stress corrosion cracking, and the problems that need to improve are also present. Ridging is a kind of surface defect that occurs as metal products are fabricated. In general, ridging causes disfigurement of the surface and a grinding process is required for amelioration. Ridging is partly caused by the large columnar structures formed during solidification [1–3]. During continuous casting of ferritic stainless steel, the equiaxed ratio of slab is normally kept in 20–30%. In the factory production process, surface ridging defect is inhibited by stimulating recrystallization during hot rolling or repeated cycles of cold rolling and annealing.

In order to improve this defect, some methods have been attempted to promote columnar-to-equiaxed transition (CET) during solidification to increase the central equiaxed area, such as low superheat casting, electro-magnetic stirring(EMS), and adding of nucleation agent [4–8]. Recently, due to the similarity of TiN and ferrite on the lattice structure, the control of TiN is considered to be an effective method for improving the equiaxed crystal ratio [9,10]. Fu [11] shows that accumulation of TiN precipitates in the microstructure in the sample central section resulted from the redistribution of Ti and N during solidification. The EBSD results show that Kurdjumov–Sachs orientation relationship

and six other ones exist in the solidified samples, which indicated ferrite could nucleate on the surface of TiN in large quantities. What's more, some oxides could act as a heterogeneous nucleation sites for TiN, and the oxides were beneficial to the further precipitation of TiN [12–15]. However, at the same time, it cannot be ignored that poor mechanical properties occur in high Ti content, affecting the subsequent processing of steel [7]. The production shows that once the size of TiN particles in steel used as a surface sheet is greater than 5 μm, the point defect will occur on the sheet surface. The content of Ti and N in steel should be strictly controlled to inhibit excessive growth of TiN particles in stainless steel.

To solve this contradiction, it is significant to employ composite addition instead of single addition. The present study proposed a magnesium–titanium treatment method for ferritic stainless steel to obtain refined TiN particles and solidification microstructure. The refining mechanism was analyzed based on classic heterogeneous nucleation theory. The research results will provide an effective guide for application of the Mg–Ti treatment technique in ferritic stainless steel.

## 2. Materials and Methods

The experiment was carried out in the resistance furnace as shown in Figure 1. To avoid oxidation of molten steel in furnace, the inert gas argon with flowrate of 1 L/min was blown from the top and bottom of the furnace. The SUS430 steel was melted in an $Al_2O_3$ crucible. As the temperature reached 1570 °C, the molten steel was held for 20 min. After the sample was completely melted, aluminum blocks, nickel–magnesium alloy (20% Mg and 80% Ni) and titanium–iron alloy (10% Ti and 90% Fe) were added in order and then stirred uniformly with a quartz rod. After that, the cylindrical quartz crucibles with a diameter of 50 mm were used to extract samples from the bath for water quenching. The remains were naturally cooled to room temperature in the furnace at the cooling rate of 0.02 °C/s. The chemical compositions with different treatments are shown in Table 1. The mass fractions of C, Si, Mn, Ti and other elements were measured by Spectro-Lab analyzer (NCS Inc., Beijing, China), and that of N was determined by ONH-3000 oxygen and nitrogen analyzer (NCS Inc., Beijing, China). In addition, the content of Mg was gauged by thermoelectric 6300 ICP spectrometer (Varian Medical Systems, UT, USA).

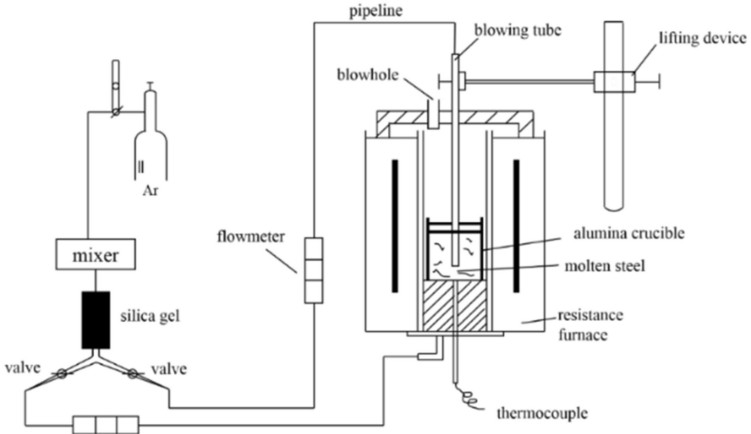

**Figure 1.** Diagram of experimental apparatus.

**Table 1.** Chemical composition of SUS430 (wt%).

| Element | C | Si | Mn | Ni | Cr | Ti | Mg | N |
|---|---|---|---|---|---|---|---|---|
| Sample 1 | 0.05 | 0.41 | 0.45 | 0.30 | 16.4 | 0 | 0 | 0.040 |
| Sample 2 | 0.06 | 0.46 | 0.41 | 0.38 | 15.7 | 0.05 | 0 | 0.053 |
| Sample 3 | 0.06 | 0.45 | 0.43 | 0.39 | 16.1 | 0.05 | 0.0017 | 0.048 |
| Sample 4 | 0.05 | 0.47 | 0.45 | 0.35 | 16.5 | 0.12 | 0 | 0.056 |

The samples were etched by aqueous hydrochloric acid solution in 70 °C. The equiaxial crystal ratio was determined by equiaxial crystal length ratio to the diameter of the cross-section area. The large crystal along the outer side was regarded as columnar crystal. Every sample was measured morn than 20 different lines cross the center point. The grain features were observed by ZEISS Axio Vert metallographic microscope (Carl Zeiss Inc., Jena, Germany). The grain size was counted by IPP image analysis software (Image Pro Plus 6.0, Media Cybernetics Inc., MD, USA). The hydrochloric acid solution was centrifuged by HC-2518 centrifuge (Zonkia Inc., Anhui, China) repeatedly after corrosion with rotating speed of 10,000 r/min. Finally, TiN was extracted, dried and transferred to a silicon wafer. The morphology, characteristics and composition of TiN were observed by SU5000 scanning electron microscope with EDS analyzer (Hitachi Inc., Tokyo, Japan). The acceleration voltage was 30 kV and the scan rate was 20 mm/s. In order to obtain clear interface morphology among $MgAl_2O_4$, TiN and matrix, TEM samples of the complex inclusion with thickness less than 50 nm were obtained by focused ion beam (FIB, HELIOS NanoLab 600i, FEI Company, Hillsboro, OR, USA), and composition distribution, micro morphology and diffraction patterns were studied by high resolution transmission electron microscopy (HRTEM, FEI TalosF-200x, accelerating voltage 200 kV, point resolution 0.25 nm, STEM resolution 0.16 nm, FEI Inc., OR, USA).

## 3. Results

### 3.1. Solidification Structure

The Ti content in samples increased from 0% to 0.12% in this experiment. In order to compare the effect of Mg content on refinement of the microstructure, Mg element with 17 ppm (wt) was added on the base of 0.05% Ti content. Figure 2 shows the solidification structures of the samples under different Ti and Mg contents, the sample diameter is 50 mm and is cooled in furnace. Comparing the solidification structures, it was found that as the Ti content in steel increased, the structure refined obviously. Under the same Ti content, addition of Mg element is beneficial to refine the microstructure further. Analyzed accurately by software IPP image, the result is shown in Figure 3a. With the addition of 0.05% Ti, the equiaxed crystal ratio increased from 15% to 37%, and the mean grain size also decreased from 4.24 mm to 1.79 mm shown as Figure 3b. Further increase of Ti content to 0.12%, the equiaxial crystal ratio increased to 48%. The grain refinement was related to the heterogeneous nucleation behavior of δ-Fe with TiN which is precipitated at the solidification front [16–18]. The mechanism of the heterogeneous nucleation behavior remains to be further explored.

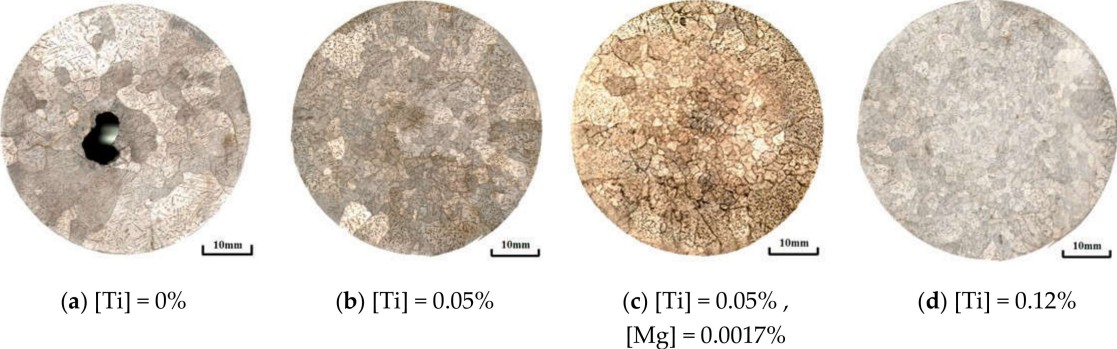

(**a**) [Ti] = 0%       (**b**) [Ti] = 0.05%       (**c**) [Ti] = 0.05% , [Mg] = 0.0017%       (**d**) [Ti] = 0.12%

**Figure 2.** Effect of Ti and Mg content in steel on solidification structure: (**a**) [Ti] = 0%; (**b**) [Ti] = 0.05%; (**c**) [Ti] = 0.05%, [Mg] = 0.0017%; (**d**) [Ti] = 0.12%.

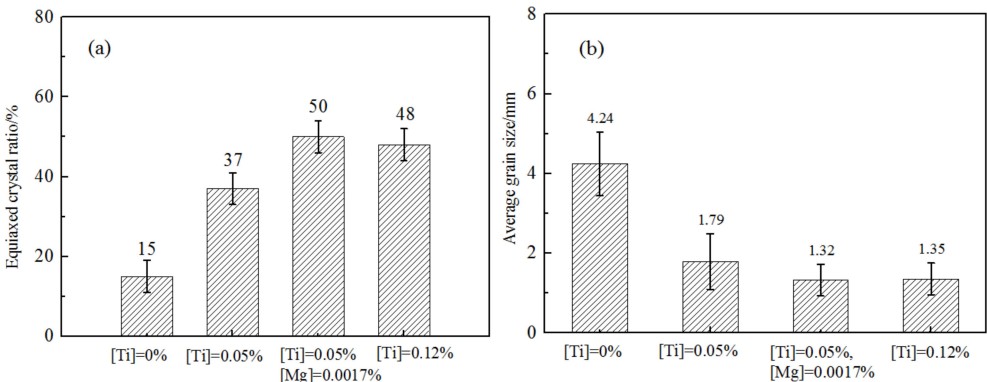

**Figure 3.** The effect of Ti and Mg contents on the equiaxial crystal ratio and average grain size:
(**a**) equiaxial crystal ratio; (**b**) average grain size.

The effect of Mg element on solidification structures can be seen as Figure 2a,b. As the Ti content was stable as 0.05%, with the addition of 0.0017% Mg element, the equiaxed crystal ratio increased from 37% to 50% and mean grain size decreased from 1.79 mm to 1.32 mm. It is obvious that the addition of Mg element was more beneficial to the formation of equiaxed grains. In order to explore the reason, the characteristics of inclusions in microscopic are also required to be investigated.

### 3.2. The Characteristic and Composition of TiN Inclusions

The grain refining was closely related to the number density of TiN. The morphology of the extracted TiN is shown in Figure 4. The typical shape of TiN was a cube and the size of TiN particles became smaller after Mg addition under the same Ti content. During the experiment, more than 40 particles were measured for comparing the size distribution of TiN particle in samples. The size distribution is shown as Figure 5; the TiN particle size in the sample before Mg addition changed from 4.2 μm to 5.8 μm and the mean value was about 5 μm. After adding the Mg element, the TiN size decreased obviously as shown in Figure 5b and it changed from 1.2 μm to 2.8 μm. The mean size was about 1.6 μm which was reduced by 60% comparing that without Mg.

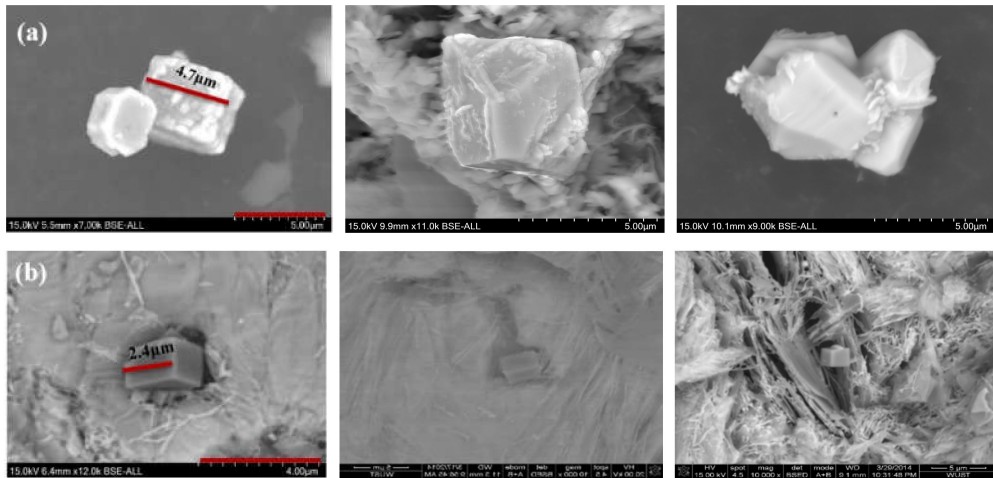

**Figure 4.** Three-dimensional morphology of typical TiN inclusions: (**a**) Mg free, (**b**) Mg addition.

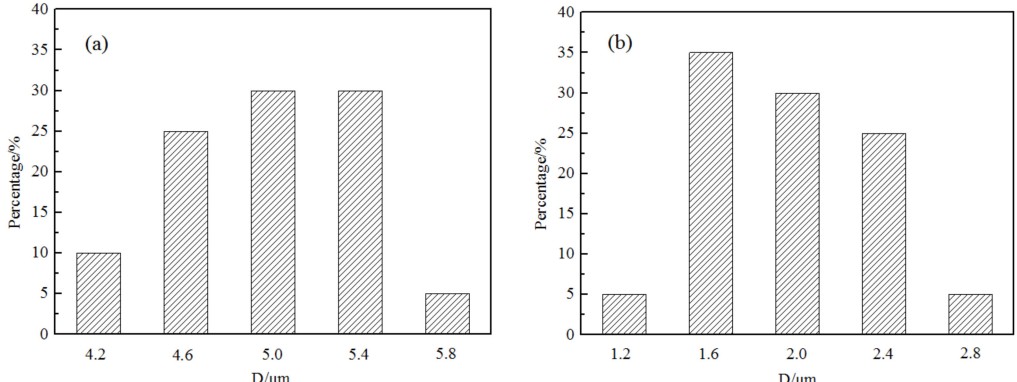

**Figure 5.** Size distribution of TiN in samples before and after adding Mg: (**a**) Mg free, (**b**) Mg addition.

The composition of two typical TiN particles in samples after Mg addition were revealed by map scanning of SEM-EDS, which are shown in Figures 6 and 7. The brightness of the color represents the content of the corresponding element. It can be observed that the Ti and N elements were uniformly distributed in the complex particle except the black core in the center. Similarly, the element in the black core was Mg and O in Figure 6 or Al, Mg and O in Figure 7. Through calculating the atomic ratio by the element content ratio on the core, Mg:Al = 2:1 was obtained. The black cores can be identified to be MgO or $MgAl_2O_4$, the grey part to be TiN. $MgAl_2O_4$ was formed by interfacial reaction at the interface of TiN/MgO [19–22]. MgO or $MgAl_2O_4$ increased the nucleation ratio of TiN at the beginning of solidification and inhibited the growth of TiN, as a result, the composite particles were formed at a relatively small size. The formation mechanism of complex particles and nucleation effect are discussed in the following part.

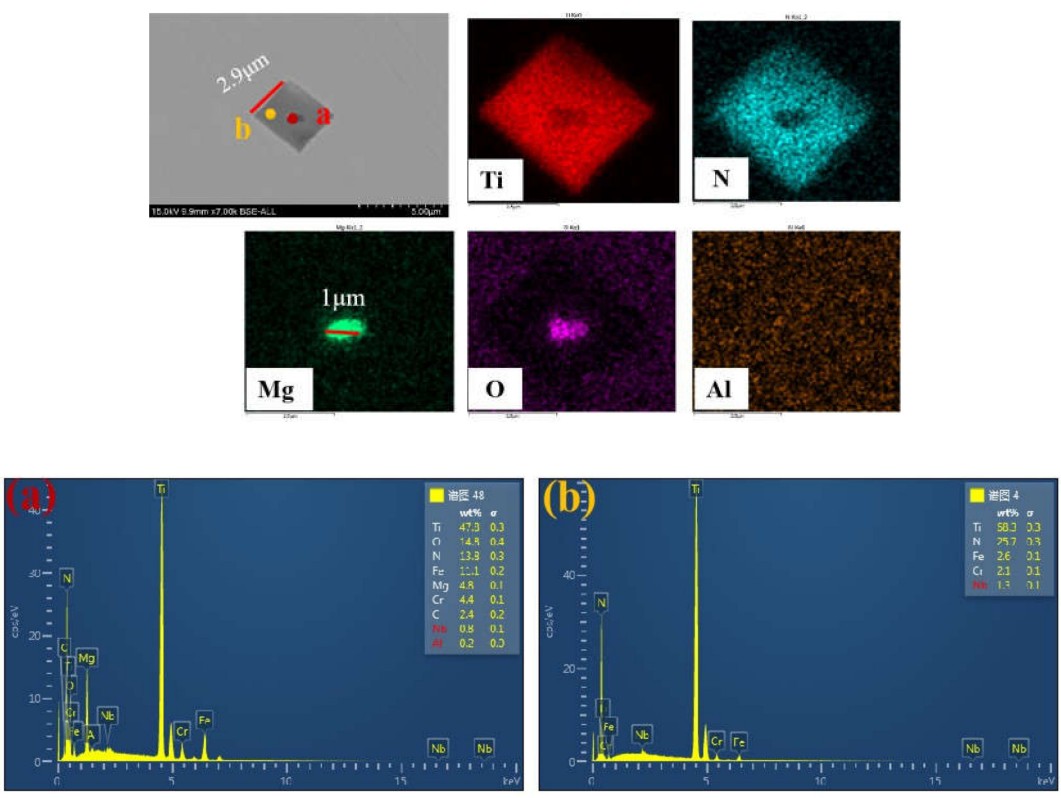

**Figure 6.** Characteristics of MgO–TiN particles in sample with Mg addition. (**a**) the oxides core; (**b**) TiN particle.

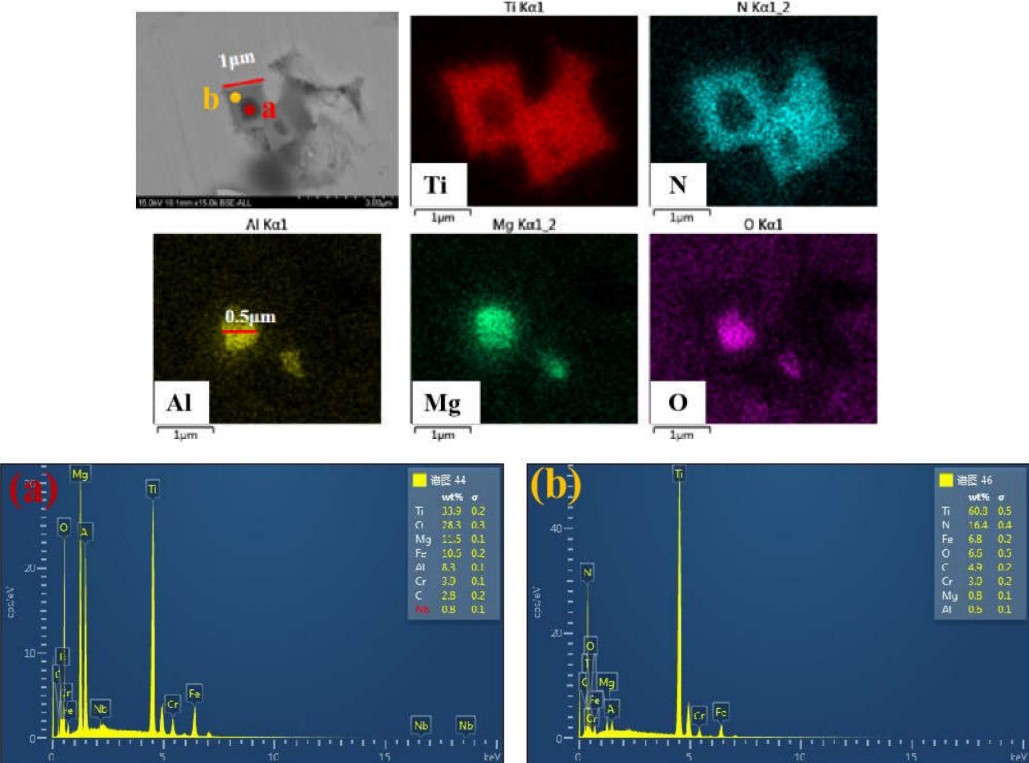

**Figure 7.** Characteristics of Al$_2$O$_3$-MgO-TiN particles in sample with Mg addition. (**a**) the oxides core; (**b**) TiN particle.

### 3.3. The Formation Mechanism of Complex Nucleus

As mentioned above, addition of Mg in ferritic stainless steel improved the refinement of grains and TiN particles. This can be attributed to enhanced heterogeneous nucleation. Mg of high vapor pressure yielded a large number of Mg gas bubbles in molten pool, which reacted with dissolved oxygen and formed deoxidation MgO particles at nanometer scale during rising of gas bubbles. These particles became cores of subsequent heterogeneous nucleation. Figure 8 shows a TEM-EDS image of a complex inclusion, which clearly comprised three layers. The core was MgAl$_2$O$_4$, the intermediate layer was TiN and outer layer was (Ti, Nb)C. The three-layer structure indicates that MgAl$_2$O$_4$ was formed at the earliest stage during solidification and was a deoxidation product. The TiN size was about 3.5 μm and it meant that the intermediate layer was formed earlier than complete solidification. The thickness of the third layer was less than 0.5 μm and TiC and NbC were both precipitated at relatively low temperature. The precipitation sequence of these three fully confirmed the heterogeneous nucleation in Mg addition stainless steel. It is well known that interfacial energy is a critical criterion to measure interface stability, which has important significance for increasing material properties. In general, the atomic arrangement along an interface, such as coherent or semi-coherent lattice relationship, will enhance the stability of interface. According to lattice disregistry, atomic arrangement is able to be determined.

To understand the grain refining mechanism of Mg grain refiner, we focused on the MgAl$_2$O$_4$/TiN interface and the TiN/δ interface. Figure 9 shows a TEM bright field image of complex inclusion particles and the selected area electron diffraction (SAED) patterns taken from MgAl$_2$O$_4$, TiN and matrix. It is noted that the intermediate layer consisted of TiN, TiC and NbC, whose crystal structures were FCC and lattice parameters had no significant difference [23–25]. The SAED patterns of TiN and (Ti, Nb)C also keep consistent in the present analysis, therefore we will mainly discuss lattice matching characteristics among MgAl$_2$O$_4$, TiN and matrix (δ-Fe) in the following text. The crystal indices are labeled in Figure 8. Figure 10 shows the overlapped SAED patterns of MgAl$_2$O$_4$/TiN and

TiN/δ. The bright spot means good lattice matching at a certain crystal plane. It can be seen from Figure 10a that (400)MgAl$_2$O$_4$ and (200)TiN matched well. This indicates that MgAl$_2$O$_4$ could act as good refiner for TiN particles. Likewise, as shown in Figure 10b, (200)TiN and (110)δ-Fe also had good matching characteristics. Furthermore, refined TiN particles were very effective in promoting heterogeneous nucleation of matrix.

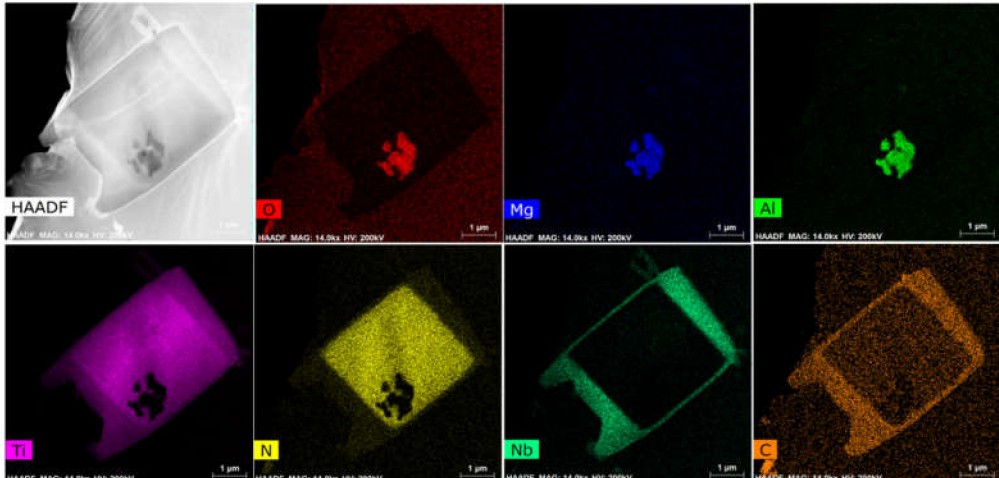

**Figure 8.** TEM-EDS image of inclusion in Mg added sample.

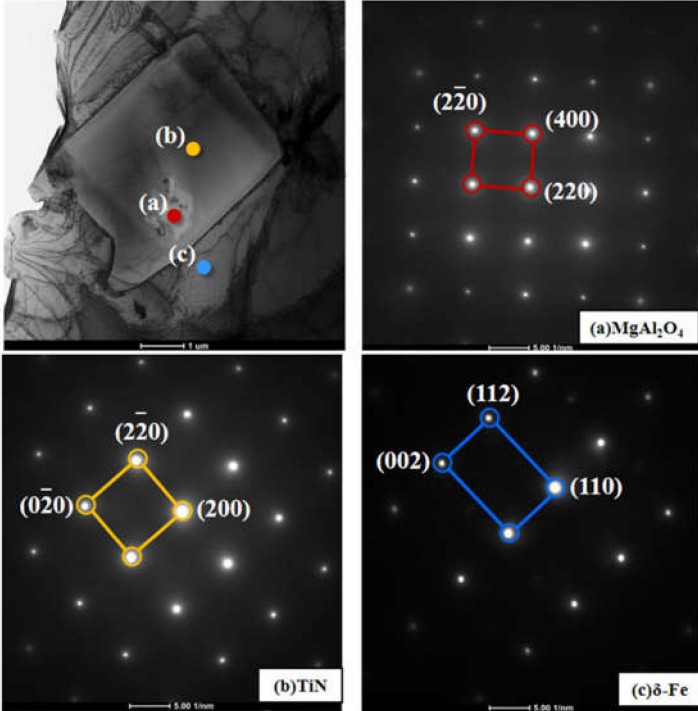

**Figure 9.** Selected area electron diffraction (SAED) patterns taken from MgAl$_2$O$_4$, TiN and δ.

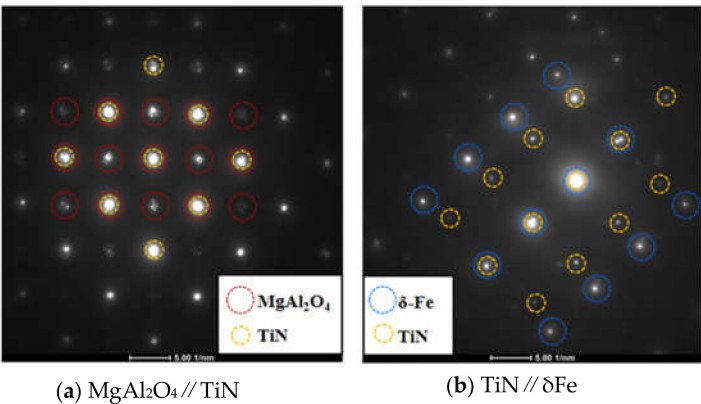

(**a**) MgAl₂O₄ ∥ TiN    (**b**) TiN ∥ δFe

**Figure 10.** Overlapped SEAD pattern between adjacent phases. (**a**) MgAl₂O₄ and TiN; (**b**) TiN and δFe.

## 4. Discussion

### 4.1. The Nucleation Effect

According to the interface coherence theory, when the lattice structure of heterogeneous phase similar to that of the melt and the interface between the heterogeneous phase and melt has a low surface free energy [26], the heterogeneous phase can effectively promote nucleation of melt. In order to measure the effectiveness of heterogeneous nucleation, Bramfitt [27,28] proposed the concept of lattice misfit to measure the effectiveness of heterogeneous nucleation. The formula is as follows:

$$\delta^{(hkl)_s}_{(hkl)_n} = \sum_{i=1}^{3} \left[ \frac{\left| d^i_{[uvw]s} \cos\theta - d^i_{[uvw]n} \right|}{d^i_{[uvw]n}} \right] \times \frac{1}{3} \times 100\%$$

where $d[uvw]s$, $d[uvw]n$ are the interatomic spacings along direction $[uvw]s$, $[uvw]n$ and $\theta$ is the angle between $[uvw]s$ and $[uvw]n$. The subscripts $s$ and $n$ stand for substrate and nucleated crystal, respectively. The lattice parameters [29] of the relevant material are given in Table 2. The (100) plane is selected as the matching crystal plane. The orientation relationship is shown in Figure 11. According to Bramfitt's calculation model, the calculation results are shown in Table 3. The disregistry between MgO and TiN was 0.068 and that between MgAl₂O₄ and TiN was 5.02. In addition, the disregistry between TiN and δ-Fe was 4.41. A disregistry less than 12% belongs to the effective nucleation range. It can be concluded that MgO and MgAl₂O₄ can promote nucleation of TiN, and TiN can also serve as the nucleation substrate of δ-Fe. So far, there is a bit of evidence to demonstrate or to interpret the effect of lattice misfit on heterogeneous nucleation behavior of liquid metal with regard to the original discussions on the relationship between misfit and undercooling [3,30,31].

**Table 2.** Parameters of the substrate and nucleation phase.

| Substance | Structure | Lattice Constant/(10⁻⁸ cm) | Lattice Constant (1773 K)/(10⁻⁸ cm) |
|---|---|---|---|
| TiN | fcc | 4.2419 | 4.3055 |
| δ-Fe | bcc | 2.8664 | 2.9396 |
| MgO | fcc | 4.216 | 4.302 |
| MgAl₂O₄ | fcc | 8.0887 | 8.1778 |

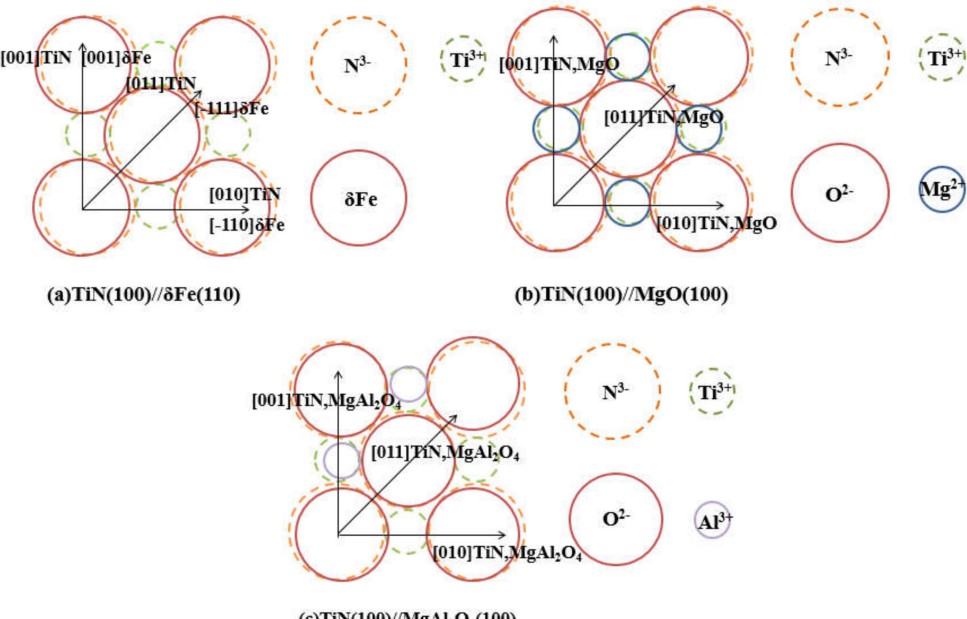

**Figure 11.** The crystallographic relationship at the interface between (100)TiN and (110) δ-Fe (**a**), between (100)TiN and (100)MgO (**b**), and between (100)TiN and (100) $MgAl_2O_4$ (**c**).

**Table 3.** Results of lattice misfit.

| Case | $[uvw]_s$ | $[uvw]_n$ | $d[uvw]_s$ | $d[uvw]_n$ | θ(deg) | δ (%) |
|---|---|---|---|---|---|---|
| | [001] | [−111] | 4.305 | 4.156 | 45 | |
| (100)TiN//(110)δ-Fe | [011] | [001] | 3.044 | 2.939 | 45 | 4.41 |
| | [010] | [−110] | 4.305 | 4.156 | 0 | |
| | [001] | [001] | 4.302 | 4.305 | 0 | |
| (100)MgO//(100)TiN | [011] | [011] | 3.042 | 3.044 | 0 | 0.068 |
| | [010] | [010] | 4.302 | 4.305 | 0 | |
| | [001] | [001] | 4.089 | 4.305 | 0 | |
| (100)$MgAl_2O_4$//(100)TiN | [011] | [011] | 2.891 | 3.044 | 0 | 5.02 |
| | [010] | [010] | 4.089 | 4.305 | 0 | |

### 4.2. Relationship between Undercooling and TiN Formation

Undercooling is a necessary condition for nucleation of melt. The promotion of heterogeneous nucleation is inseparable from the analysis of nucleation supercooling. The schematic describes the process that equiaxed crystals nucleated on the surface of TiN with core, shown in Figure 12. $T_a$ is the temperature gradient due to the heat flux, and $T_L$ is the equilibrium liquidus temperature curve. Before the solidification interface, $T_a$ intersects with $T_L$. The constitutional undercooling region caused by the redistribution of the solute of the advancing solid/liquid interface is generated before the interface. The equiaxed crystal is easily formed when the maximum of the constitutional undercooling is greater than that required for the heterogeneous nucleation in broad undercooling conditions [32–36].

The nucleation of the crystal needs to occur under the generation of sufficient undercooling. The undercooling (ΔT) is related to the melt itself and the characteristic and number density of heterogeneous nucleon particles in the melt. The presence of MgO and $MgAl_2O_4$ in the molten steel makes TiN easier to nucleate and increase the nucleation ratio. When the temperature is lowered to its precipitation temperature, a large number of TiN partilces with smaller size are obtained, which distributed uniformly makes the maximum of constitutional undercooling before the solid–liquid interface exceeding the subcooling of heterogeneous nucleation. A new nucleation process has occurred,

which has changed from the exogenous growth of columnar dendrites to the endogenous growth of equiaxed dendrites, increasing the equiaxed crystal ratio and refining the solidified structure.

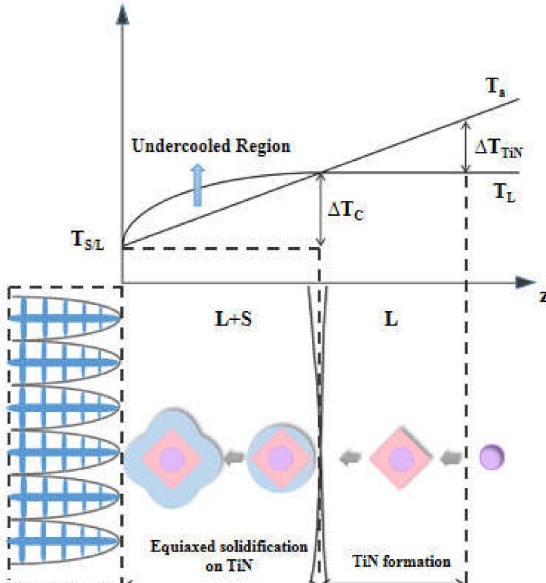

**Figure 12.** Diagram of the nucleation of an equiaxed crystal on a TiN surface with an oxide core.

## 5. Conclusions

In the present study, Mg–Ti composite processing was proposed in ferritic stainless steel. The effect of Mg–Ti treatment on refinement of solidification structure and TiN particles was investigated. The refinement mechanism was discussed by interface coherence theory. It came to the following conclusions.

(1) Addition of Ti element is beneficial to refining microstructure. The equiaxed crystal ratio increased from 15% to 37% after adding only 0.05% Ti. Based on this, the equiaxed crystal ratio was improved from 37% to 50% with an additional 17 ppm of Mg element. TiN particles could promote nucleation of the $\delta$-Fe phase to refine the ferrite structure. After the addition of Mg element, the size of TiN particles decreased. Moreover, TiN had a core of MgO or $MgAl_2O_4$ in this case, which promoted formation of TiN particles.

(2) The crystal relationships is $(200)TiN//(110)\delta$-Fe and $(400)MgAl_2O_4//(200)TiN$ among TiN $\delta$-Fe and $MgAl_2O_4$. According to the interface coherence theory, the disregistry between MgO and TiN is 0.068 and that between $MgAl_2O_4$ and TiN is 5.02. Similarly, the disregistry between TiN and $\delta$-Fe is 4.41. All of them belong to the effective nucleation range.

(3) The presence of MgO and $MgAl_2O_4$ in the molten steel enhanced the nucleation of TiN particles. A large amount of TiN prticles begin to formed which distributed uniformly produced the maximum of constitutional undercooling before the solid–liquid interface exceeding the subcooling of heterogeneous nucleation. A new nucleation has occurred which inhibits the growth of the columnar crystal and enhanced the equiaxed crystal ratio.

**Author Contributions:** Conceptualization, D.W.; methodology, T.Q.; software, J.Z.; validation, J.T., and D.H.; formal analysis, T.Q.; investigation, C.Z.; resources, C.Z.; data curation, J.Z.; writing—original draft preparation, C.Z.; writing—review and editing, T.Q.; visualization, J.T.; supervision, D.W.; project administration, D.W.; funding acquisition, D.W. All authors have read and agreed to the published version of the manuscript.

**Funding:** This research was funded by the National Natural Science Foundation of China (Grant No. 51774208, 51674172 and 51704200).

**Conflicts of Interest:** The authors declare no conflict of interest.

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
