# Peer review of "Effect of Mg–Ti Treatment on Nucleation Mechanism of TiN Inclusions and Ferrite"

_metals, doi:10.3390/met10060755_

Round 1
Reviewer 1 Report
It is a good, interesting and well written paper, deserves to be published as it is
Author Response
Dear reviewer,
Thanks for your comments.The manuscript has been improved.
Reviewer 2 Report
74-76, How many samples are used to detrmination of Grain Size?
76, explanation about which methodes are used to measure the equiaxial crystal ratio?
91- is it just a coincidence or did you find same result in different area?
110-115- Definition of methods, which used in this Investigation/Figure for example SEM- or TEM-Method?
173- There is insufficient to describe Figure 11, more detail would be better.
181- It would be better to say to “Ta” as “temperature gradient due to the heat flux”
Reviewer 3 Report
-Very interesting results on the refinement of steel
-Good approach to nucleation theory and Bramfitt model
-The study was well put together
-Significant improvement of English is required. In particular, grammar must be improved.
-Authors must be diligent to follow guidelines of measurement units. For example, spaces after number and unit is required, volts are denoted with a capital V etc.
-The introduction did not appear to be exhaustive enough given the topic. I would expect more examples of current literature. A better review of what has been attempted in this particular field would be beneficial. Furthermore, a critical analysis of the shortcomings of current research and areas for improvement is a must.
-The impact of the study as it relates to the steel industry and academia as a whole is required.
-How was the crystal ratio determined in this study?
-What were the number of samples/trials used to determine the grain size and crystal ratio?
-The authors repeatedly mention nucleation ratio in the article. Please explain the definition of nucleation ratio as it is not a familiar term. The reviewer believes that this term may be confused with nucleation rate.
-The authors state that MgO both promotes nucleation and inhibits growth of the resulting grains. Please explain this phenomenon further.
-Overall, the figure caption must be improved. For examples, Figure 2, the caption does not describe each case i.e. a, b, c
-Axis labels are missing for figure 3
-Figure 5 units are missing on the ordinate
-In table 3, the authors state that there is a 45 degree offset from the closed packed directions of Fe and TiN. However, on the images presented in figure 11, no offset/angle can be see in the direction. Specifically, the [010] for TiN appears to be parallel to [1-10] for Fe. This is a large discrepancy and must be revisited. If there is indeed a 45 degree angle between the directions, separate vectors should be used such that the offset is clearly visible.
-Also for table 3, the third row, (100)MA//(100)TiN, what is MA?
-Section 4.2, although it appears that the authors have a good grasp of the solidification/nucleation theory, a significant lack of citations is apparent. The review cannot believe that this section consists of original ideas from the authors as sections of Gruzleski textbooks come to mind.
